# Evaluation of a Commercial Point-of-Care RT-LAMP Assay for Rapid Detection of SARS-CoV-2

**DOI:** 10.3390/biomedicines11092344

**Published:** 2023-08-23

**Authors:** Janet Hei Yin Law, Wai Sing Chan, Tsun Leung Chan, Edmond Shiu Kwan Ma, Bone Siu Fai Tang

**Affiliations:** Department of Pathology, Hong Kong Sanatorium and Hospital, Hong Kong SAR, China; janethy121@gmail.com (J.H.Y.L.); waising.chan@connect.polyu.hk (W.S.C.); chris.tl.chan@hksh.com (T.L.C.); eskma@hksh.com (E.S.K.M.)

**Keywords:** COVID-19, Detect COVID-19 Test, RT-LAMP, POCT, SARS-CoV-2, variants of concern

## Abstract

The goal of this study was to evaluate the performance of a commercial reverse transcription loop-mediated isothermal amplification (RT-LAMP) assay (Detect COVID-19 Test) in the detection of severe acute respiratory syndrome coronavirus 2 (SARS-CoV-2). A total of 202 human respiratory and viral culture specimens were tested retrospectively. The performance of the Detect COVID-19 Test was comparable to that of commercial real-time polymerase chain reaction assays (sensitivity: 93.42%; specificity: 100%), and better than that of the rapid antigen test (sensitivity: 48.00%; specificity: 100%) for specimens with threshold cycle (Ct) values of less than 30. The Beta, Delta, and Omicron variants of concern were successfully detected. With their simplicity of use and good assay sensitivity, point-of-care RT-LAMP assays may be a viable option for SARS-CoV-2 testing at home, or in regions without sophisticated laboratory facilities.

## 1. Introduction

It has been more than 3 years since the World Health Organization characterized the outbreak of coronavirus disease 2019 (COVID-19) as a pandemic [1]. The etiologic agent, severe acute respiratory syndrome coronavirus-2 (SARS-CoV-2), has caused nearly seven million deaths [2] since the first reported outbreak in Wuhan, China [3]. The rapid and accurate detection of the highly contagious SARS-CoV-2 is crucial to the timely delivery of optimal treatment, as well as to limiting the spread of COVID-19. Currently, there are two major methods for SARS-CoV-2 screening. The first is a reverse transcription real-time polymerase chain reaction (rRT-PCR), targeting conserved regions of the SARS-CoV-2 genome. This method has the advantages of a high sensitivity, specificity, and throughput, and yet it is mostly confined to laboratory settings, and requires sophisticated instruments and trained medical laboratory technologists [4]. An alternative to rRT-PCR is a rapid antigen test (RAT). In general, RATs are lateral flow tests that contain gold-nanoparticle-conjugated antibodies targeting the SARS-CoV-2 protein, with the nucleocapsid protein being a popular target [5]. The visualization of the target and control bands is facilitated by the aggregation of gold nanoparticles, and can be read with the naked eye. Compared to rRT-PCR, RATs do not involve complicated steps and instruments, the sample-to-result time is short, and the reagent cost is low. These are the reasons why RATs were widely adopted as a first-line point-of-care testing (POCT) method during the peak of the COVID-19 pandemic. However, the sensitivity of RATs is low and, hence, they are prone to false-negative results [6,7].

Recently, researchers have been exploring the utility of reverse transcription loop-mediated isothermal amplification (RT-LAMP) for the point-of-care detection of SARS-CoV-2 [8,9,10]. For instance, Chow and co-workers developed a small van-sized mobile COVID-19-LAMP diagnostic unit for providing an on-site SARS-CoV-2 screening service [8]. Among a cohort of respiratory and SARS-CoV-2 culture specimens, the sensitivity was estimated to be at least 93.33%. In another study by Yip and co-workers, a nsp8 one-tube RT-LAMP-CRISPR (clustered regularly interspaced short palindromic repeat) assay was developed, showing a sensitivity of 98.6%, and a specificity of 100%, for SARS-CoV-2 detection [9]. Rivas-Macho and coworkers have evaluated an extraction-free colorimetric RT-LAMP method for the detection of SARS-CoV-2 in saliva [10]. They reported that the sensitivity and specificity of the assay were 90% and 100%, respectively, for saliva specimens with a Ct value equal to or less than 34. In addition, RT-LAMP has been tested/adopted in airports for testing upon arrival/departure, with a cost in between those of the RAT and rRT-PCR [11,12]. In this study, we attempted to evaluate the performance of a commercial RT-LAMP assay, Detect COVID-19 Test (abbreviated as Detect in the following text), which is an alternative test to the nucleic acid amplification assay and RAT. It reversely transcribes and amplifies the open reading frame 1ab (ORF1ab) sequence of SARS-CoV-2 genome via LAMP. The amplified product is detected and visualized via a lateral flow device. In this study, we evaluated and compared the performance of this POCT with commercial rRT-PCR assays and RATs.

## 2. Materials and Methods

### 2.1. Collection of Specimens

A total of 202 specimens were tested retrospectively with the Detect COVID-19 Test (Detect Inc., Connecticut, Panama City, FL, USA), which included 198 collected from the human respiratory tract, 3 viral culture specimens in viral transport medium (VTM) containing SARS-CoV-2 Beta or the Delta variant of concern (VOC), and an RNA extract of the Omicron VOC (Table 1). The respiratory specimens included single or combined nasal/nasopharyngeal/throat swabs in VTM, deep-throat saliva (DTS), and lower-respiratory tract (LRT) specimens, such as bronchoalveolar lavage, sputum, and tracheal aspirate. The specimens were archived at −80 °C after routine rRT-PCR testing for SARS-CoV-2 at the Department of Pathology, Hong Kong Sanatorium & Hospital, between July 2020 and March 2022.

### 2.2. Routine rRT-PCR Assays

Xpert Xpress SARS-CoV-2 (Cepheid, Sunnyvale, CA, USA) and Simplexa COVID-19 Direct (Diasorin Molecular LLC, Cypress, CA, USA) were the primary rRT-PCR methods for routine testing. Briefly, for Xpert Xpress SARS-CoV-2, 300 µL of VTM was added directly into the specimen port, before the cartridge was loaded into the GeneXpert IV system (Cepheid, Sunnyvale, CA, USA), in accordance with the manufacturer’s recommended procedure. For DTS or LRT specimens, 500 µL of the specimen was mixed with 500 µL of Sputasol (Oxoid, Cambridge, UK) at the working concentration, until liquefication. The liquefied specimen was centrifuged at 13,000× *g* for 1 min, and 300 µL of the supernatant was used for rRT-PCR. For Simplexa COVID-19 Direct, 50 µL of VTM or the supernatant of the Sputasol-treated DTS/LRT specimen was added to the sample port of an 8-well direct amplification disc, following the manufacturer’s recommended procedure for testing with the Diasorin Liason MDX system (Diasorin Molecular LLC, Cypress, CA, USA). The RNA extract of the Omicron VOC was tested via Simplexa COVID-19 Direct only.

### 2.3. Detect COVID-19 Test

Detect was performed, with a slight modification in the specimen inoculation step. Briefly, 75 µL of the VTM/RNA extract/the supernatant of the Sputasol-treated DTS or LRT specimen was directly inoculated onto the flocked swab provided. The inoculated flocked swab was then inserted into a buffer-containing test tube, and stirred for 15 s. The test tube was closed with a screw cap containing the reagent bead; this was followed by inversion up-and-down and shaking side-to-side, until the reagent bead dissolved completely into the sample. The test tube was snapped downward a few times, to bring all the liquid to the bottom of the tube. The test tube was then inserted into the powered-up Detect Hub, accompanied by a “beep” sound and a blinking green light. The RT-LAMP lasted for 55 min, and a solid green light indicated the completion of the process. The tube was assembled into Detect Reader after the addition of the reagent. The amplified products of the SARS-CoV-2 ORF1ab and human gene control were visualized as red lines in the lateral flow strip in 10 min. Red line 1 was the SARS-CoV-2 ORF1ab, and red line 2 was the human gene control. Photographs were taken and saved in Joint Photographic Experts Group (JPEG) format for reference. The results were interpreted visually, in accordance with the manufacturer’s instructions (Figure 1). Briefly, for a positive result, red line 1 (SARS-CoV-2 ORF1ab) must be present, irrespective of the presence of a control line. For a valid negative result, red line 2 (human gene control) must be present.

### 2.4. Rapid Antigen Test

The INDICAID COVID-19 rapid antigen test (Phase Scientific, Hong Kong, China) was performed, with a slight modification in the specimen inoculation step, which was in line with Detect. The entire cap of the buffer solution vial was removed, and the inoculated swab was stirred in the buffer solution of the RAT; this was followed by the rolling of the swab against the inner wall of the vial, to release the liquid from the swab, before disposal. The buffer solution vial was capped, and the top half of the cap was removed. Three drops of specimen were added to the sample well of the lateral flow device, and the result was read visually at 20 min. The RNA extract of the Omicron VOC viral was not tested via this method.

## 3. Results

Of the 202 specimens tested, the results of 12 specimens (12/202, 5.94%) were invalid and, therefore, were excluded from the final analysis. The remaining 190 specimens were classified into four groups, primarily based on the Xpert Xpress SARS-CoV-2 results as the reference standard, with the lower threshold cycle (Ct) value being considered. For the RNA extract of the Omicron VOC, the ORF1ab Ct value of the Simplexa COVID-19 Direct assay was considered. The four groups were positive specimens with: (1) a Ct value of less than 30, (2) a Ct value ranged between 30 and 35 inclusive, (3) a Ct value greater than 35, and (4) a ‘not detected’ result. Table 2 summarizes the sensitivity and specificity of Detect and the RAT. For all specimens with a Ct value of zero (not detected, *n* = 49), both Detect and the RAT were 100% concordant with the rRT-PCR results and, therefore, the specificity was 100%. For positive specimens with a Ct value of less than 30 (76 specimens for Detect; 75 specimens for the RAT, excluding the RNA extract), the sensitivity of Detect was 93.42% (71/76), and that of the RAT was 48.00% (36/75). For specimens with a Ct value between 30 and 35 (*n* = 47), the sensitivity of Detect and the RAT dropped to 57.45% (27/47) and 6.38% (3/47), respectively. For specimens with a Ct value of greater than 35 (*n* = 18), the sensitivity of Detect and the RAT further dropped to 16.67% (3/18) and 5.56% (1/18), respectively. Table 3 shows the results of Detect for the four viral culture specimens. The results suggest that the Beta, Delta and Omicron VOCs were successfully detected.

## 4. Discussion

We are experiencing one of the largest global-scale pandemics in human history. Inn the face of the challenge of the exponential growth in infection, and of the limited resources for molecular reagents, the RAT has been an economical, rapid, and portable alternative for identifying individuals with a high viral load of SARS-CoV-2. In Hong Kong, the RAT has been adopted for regular at-home screening before school, work, conferences, and large-scale gatherings at the peak of the pandemic, which has helped to broaden the coverage of screening that would not be achievable using rRT-PCR. This notwithstanding, the sensitivity of the RAT is low. Recent studies have revealed a large variation in the overall sensitivity of different brands of RAT, ranging from 45.7 to 82.1% [6,7,13]. According to the interim guidance published by the World Health Organization, the recommended minimum performance requirements of the RAT in terms of sensitivity and specificity should be at least 80% and 97%, respectively [14]. Undoubtedly, there is still room for improvement regarding the sensitivity of this method.

According to our data, the sensitivity of the RT-LAMP assay Detect was better than the RAT among all the Ct value ranges of the positive specimens and, in particular, its sensitivity was comparable to that of rRT-PCR (93.42%) for positive specimens with Ct values of less than 30. Nevertheless, the sensitivity dropped to 57.45% and 16.67% for specimens with a lower viral load. In fact, a positive PCR result does not necessarily indicate that the individual is contagious with SARS-CoV-2. A number of studies have revealed that, at high Ct values of rRT-PCR, the infectivity was low [15,16,17]. For instance, Jaafar and coworkers investigated the viral culture results of 3790 SARS-CoV-2-quantitative-PCR-positive specimens. They observed that, at a Ct value of 25, the positive rate of the culture was about 70% and, at a Ct value of 30, the positive rate declined to 20%, and further decreased to less than 3% at a Ct value of 35 [15]. In another study by Kim and co-workers, 165 serial nasopharyngeal and oropharyngeal specimens were collected from 21 COVID-19 patients, to study the time from the illness onset to viral clearance via culturing and rRT-PCR [16]. They observed that, among the 89 specimens cultured for SARS-CoV-2, a viral culture was positive only in specimens with a Ct value of 28.4 or less. Singanayagam and coworkers attempted to culture SARS-CoV-2 from 324 upper respiratory tract specimens that tested positive via rRT-PCR, to study the relationship between the Ct value and virus isolation [17]. They observed that, at a Ct value of greater than 30 and up to 35, the culture positivity was 27.97%. On the basis of this knowledge, it could be speculated that Detect might be more sensitive than the RAT in identifying individuals who are contagious with SARS-CoV-2. In addition, our preliminary data showed that Detect might be able to cover the Beta, Delta, and Omicron VOCs. Nevertheless, as SARS-CoV-2 is constantly evolving, in silico surveillance is warranted, to assess the coverage of emerging variants.

Ideally, a low-cost, rapid (with an assay time comparable to common RATs; for instance, 15 min), reasonably sensitive, highly specific, mutation-tolerant, and easy-to-use molecular test that combines cell lysis with the reverse transcription and amplification of the pathogen nucleic acid, with unaided result visualization, in a portable device, could be an improved surrogate for the RAT. From the point of view of a user, we have observed a number of points that might improve the usage of Detect as a POCT. As a product targeting at-home testing, there will be more variation in the ability of users to adhere to the standard procedure. The Detect procedure involves a number of steps which might not be easy to follow for laypeople. For example, we observed that the reagent bead was not readily dissolved on a number of occasions, which meant more patience and time were required. If the user is not aware of this, and continues the test straight away, the performance of the assay may be jeopardized. The total assay time of Detect is more than one hour, which may be too long for a POCT. In addition, we have encountered 12 invalid results (5.94%) in this study, and the failure rate should be improved.

Last, but not the least, our study had the following limitations. The first point concerns specimen inoculation. As mentioned in the Materials and Methods section, the swabs and viral culture specimens were preserved in VTM, and the DTS specimens were pretreated with Sputasol, prior to rRT-PCR, and they were archived at −80 °C as the status quo. As both Detect and the RAT are designed for testing direct nasal cavity specimens, the effect of the specimen dilution on the assay sensitivity should be considered. We have tried to make the comparison between Detect and the RAT fair and closer to the undiluted specimens, by equaling and maximizing the amount of specimen inoculated into both swabs. On the other hand, as the specimens were diluted and liquefied, the tolerance of Detect to viscosity and PCR inhibitors might be underchallenged. The quality and quantity of the detectable SARS-CoV-2 nucleic acid and protein might also be affected by the freeze-and-thaw process. The results of this study should be confirmed in a larger cohort, with a more even distribution across the four specimen groups, including a reasonable portion of the currently prevalent SARS-CoV-2 VOCs, to further test the assay’s inclusiveness for common variants. In addition, this study was performed in a laboratory, under controlled environmental conditions, by trained medical laboratory technologists. As the goal of Detect is at-home testing, future studies at the community level, with real-life operator and environmental variation, is warranted.

## 5. Conclusions

At the time of writing this conclusion, our city and the globe have returned to normal, being able to live with the virus. It is not easy to predict when and where the next pandemic of this scale will take place, and which pathogen will cause it. If this happens, the lessons we have learnt over the past 3 years will be useful. We believe that the findings from this study, and the many from the rest of the world, will show us the right and smart way through.

## Figures and Tables

**Figure 1 biomedicines-11-02344-f001:**
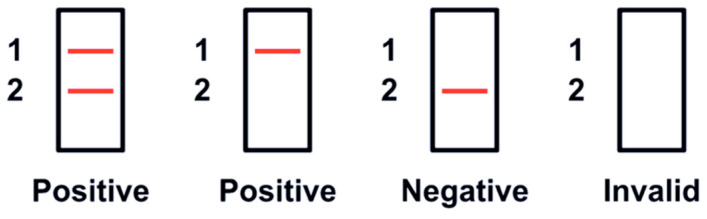
Result interpretation for the Detect COVID-19 Test.

**Table 1 biomedicines-11-02344-t001:** Specimen types included in this study.

Specimen Type	Quantity
Combined nasal and throat swabs	160
Combined nasopharyngeal and throat swabs	11
Nasal swab	1
Deep-throat saliva	22
Lower respiratory tract	4
RNA extract from viral culture	1
Viral culture specimen	3
**Total**	**202**

**Table 2 biomedicines-11-02344-t002:** Sensitivity and specificity of the Detect COVID-19 Test, and the rapid antigen test.

Ct Value	Sensitivity			Ct Value	Specificity		
	Detect	RAT	No. of Specimen		Detect	RAT	No. of Specimens
<30	93.42%	48.00%	76 *	0	100%		49
30–35	57.45%	6.38%	47				
>35	16.67%	5.56%	18				

* 75 specimens for the RAT, excluding the RNA extract from the SARS-CoV-2 viral culture. Ct, threshold cycle; RAT, rapid antigen test.

**Table 3 biomedicines-11-02344-t003:** Beta, Delta, and Omicron variants of concern detected via the Detect COVID-19 Test.

Specimen Type	VOC Type	Result of Detect	Commercial rRT-PCR Assay Results		
			Method	Target Gene(s)	Ct Value
RNA	Omicron	** 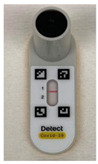 **	Simplexa COVID-19 Direct	ORF1ab	27.3
Viral culture in VTM	Delta	** 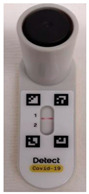 **	Xpert Xpress SARS-CoV-2	E and N genes	E: 39.9; N2: 38.6
			Simplexa COVID-19 Direct	ORF1ab	32.3
Viral culture in VTM	Delta	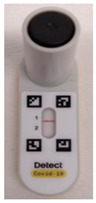	Xpert Xpress SARS-CoV-2	E and N genes	E: 34.3; N2: 35.7
			Simplexa COVID-19 Direct	ORF1ab	32.1
Viral culture in VTM	Beta	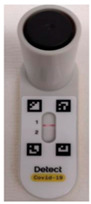			
			Xpert Xpress SARS-CoV-2Simplexa COVID-19 Direct	E and N genes ORF1ab	E: 31.6; N2: 34.030.9

Ct, threshold cycle; E, envelop; N, nucleocapsid; ORF1ab, open reading frame 1ab; VOC, variant of concern; VTM, viral transport medium.

## Data Availability

The data presented in this study are available from the corresponding author.

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
