# Peer review of "Evaluation of a Commercial Point-of-Care RT-LAMP Assay for Rapid Detection of SARS-CoV-2"

_biomedicines, 2023, doi:10.3390/biomedicines11092344_

Round 1
Reviewer 1 Report (Previous Reviewer 2)
This study described evaluation of a commercial RT-LAMP assay for detecting SARS-CoV-2 by comparison with RT-PCR and a rapid antigen test. This comparison is useful for understanding the performance of different tests. The revised manuscript has addressed previous concerns properly. I have no further suggestions.
Author Response
We would like to thank the reviewer for the precious time and comments on our work.
Reviewer 2 Report (New Reviewer)
In my opinion, the work of Janet Hei Yin Law with co-authors is very important and interesting. It is important because, despite the decrease in the incidence rate due to COVID-19, no one guarantees that the virus will not cause a new increase in cases. It is interesting because the authors attempted to develop a test concept combining PCR approaches and immunochromatographic tests. I think the authors succeeded.
The work contains a correct statistical analysis. Necessary illustrations and tables. The authors take a critical approach to the results obtained. There are minor errors in the text, but the authors, it seems to me, can easily correct them.
Reviewer 3 Report (New Reviewer)
The authors reported that the performance of the Detect COVID-19 Test was comparable to that of commercial real-time polymerase chain reaction assays and better than rapid antigen tests for specimens with threshold cycle (Ct) values of less than 30. This information can be useful as it enhances the diagnostic and screening capabilities for SARS-CoV-2 infection.
Only a few considerations: 1) In which settings could this method be applied? How cost-effective can it be on a large scale, estimating time and money savings in screening procedures? 2) Do you believe that the sensitivity and specificity of this method could vary based on current and future VOCs?
Average english quality
Author Response
We would like to thank the reviewer for the precious time and comments on our work.
Reviewer's comment 1:
In which settings could this method be applied? How cost-effective can it be on a large scale, estimating time and money savings in screening procedures?
Author response to comment 1:
In addition to at-home testing, this method may be applied at, for instance, airports for arrival/ departure testing. For instance, RT-LAMP has been adopted/ tested in airports during the pandemic. The cost and performance of RT-LAMP are between RAT and rRT-PCR. We have added the information to line 50-51 of Introduction section.
Reviewer's comment 2:
Do you believe that the sensitivity and specificity of this method could vary based on current and future VOCs?
Author response to comment 2:
We agree that both parameters may vary with emerging SARS-CoV-2 variants and hence we added the sentence ‘Nevertheless, as SARS-CoV-2 is constantly evolving, in silico surveillance is warranted to assess the coverage for emerging variants’ in Discussion section, line 197-199.
Reviewer's comment 3:
Minor editing of English language required.
Author response to comment 3:
We have proofread the manuscript and highlighted all the changes in yellow.
This manuscript is a resubmission of an earlier submission. The following is a list of the peer review reports and author responses from that submission.
Round 1
Reviewer 1 Report
Comments:
Law et al. report a study of POCT commercial LAMP detection kit called Detect Covid-19 TestTM in SARS-CoV-2 nucleic acid detection, including Beta, Delta, and Omicron variant of concerns (VOCs). They used 202 samples and found comparable results to the conventional 9 RT-PCR approach, processing a high sensitivity of 93.42% in detecting SARS-CoV-2 nucleic acid in patient samples which is much more sensitive compared to rapid antigen tests (47.56%) regarding to high viral load samples (Ct<30).
-This communication paper can be published after minor minor spell check.
Author Response
Thank you for your review. We have carefully done spell check and highlighted the changes in the manuscript. “Details are summarized in Table 3“ (Line 101). “Therefore, the Detect Covid-19 Test™ kit exhibited high sensitivity for identifying contagious individuals…”(Line 111-112). “The kit is effective in detecting contagious….”(Line 142).
Reviewer 2 Report
This manuscript evaluated a POCT commercial LAMP detection kit called Detect Covid-19 Test for SARS-CoV-2 nucleic acid detection. Although the topic is interesting, but the experimental design is poor. First, fresh samples instead of archived samples from same group of patients should be used for fair comparison. Second, more details should be given on how the Ct values of the samples were obtained, such as how much volumes of the samples was used for RNA extraction and how much was added as the templates? Third, from the results in Table 3, it is clear that the RT-PCR test is much more sensitive for detecting real samples than the DETECT COVID-19 Test. It is not right to conclude that the COVID-19 test has comparable sensitivity (line 131). From the possible limitations given in the discussion, the authors knew that there are some problems in their experimental design. Because these flaws could affect the conclusions, it is not appropriate for publication.
Author Response
Please see the attachment. We truly hope that the revised version is suitable for publication now. Thank you.
